# Pictorial Review of MRI Findings in Acute Neck Infections in Children

**DOI:** 10.3390/children10060967

**Published:** 2023-05-29

**Authors:** Janne Nurminen, Jaakko Heikkinen, Tatu Happonen, Mikko Nyman, Aapo Sirén, Jari-Pekka Vierula, Jarno Velhonoja, Heikki Irjala, Tero Soukka, Lauri Ivaska, Kimmo Mattila, Jussi Hirvonen

**Affiliations:** 1Department of Radiology, University of Turku and Turku University Hospital, 20520 Turku, Finland; jaheik4@gmail.com (J.H.); tatu.j.happonen@utu.fi (T.H.); mikko.nyman@varha.fi (M.N.); aapo.siren@varha.fi (A.S.); jari-pekka.vierula@varha.fi (J.-P.V.); kimmo.mattila@varha.fi (K.M.); jussi.hirvonen@utu.fi (J.H.); 2Department of Otorhinolaryngology-Head and Neck Surgery, University of Turku and Turku University Hospital, 20520 Turku, Finland; jarno.velhonoja@varha.fi (J.V.); heikki.irjala@varha.fi (H.I.); 3Department of Oral and Maxillofacial Surgery, University of Turku, 20014 Turku, Finland; tero.soukka@varha.fi; 4Department of Paediatrics and Adolescent Medicine, InFLAMES Research Flagship Center, University of Turku and Turku University Hospital, 20520 Turku, Finland; lauri.ivaska@varha.fi

**Keywords:** magnetic resonance imaging, infection, neck, emergency medicine, pediatric

## Abstract

Pediatric neck infections and their complications, such as abscesses extending to deep neck compartments, are potentially life-threatening acute conditions. Medical imaging aims to verify abscesses and their extensions and exclude other complications. Magnetic resonance imaging (MRI) has proven to be a useful and highly accurate imaging method in acute neck infections in children. Children and adults differ in terms of the types of acute infections and the anatomy and function of the neck. This pictorial review summarizes typical findings in pediatric patients with neck infections and discusses some difficulties related to image interpretation.

## 1. Introduction

Most conventional neck infections can be managed conservatively. However, deep neck infections possess a high risk for serious complications, such as abscess formation, mediastinitis, airway compromise, and venous thrombosis due to the infection spreading along the fascial planes and potential face and neck spaces. These complications are potentially lethal [1]. In children, prompt surgical treatment may be necessary, including cervical incision, intraoral incision, immediate tonsillectomy, or rarely, dental extraction [2]. Large abscesses and a younger age seem to predict surgical intervention [3,4].

Pediatric acute neck infections arise from common infections originating in the ears, nose, or throat and have the potential to disseminate to the deep neck spaces through direct continuity or lymphatic drainage to the lymph nodes [5]. When adequate cultivation procedures are utilized, anaerobic bacteria can be identified from most abscesses. Untreated abscesses can spontaneously rupture into the pharynx, resulting in fatal aspiration [6]. Neck swelling can initially be imaged with ultrasound, which has a high resolution for superficial structures. The main indication for cross-sectional imaging (as elaborated below) is the suspicion of a surgically drainable abscess in deep neck spaces for which ultrasound is insufficient.

In clinical practice, these infections have typically been imaged with contrast-enhanced computed tomography (CECT) [7,8], which has limited ability to differentiate abscess from lymphadenitis, cellulitis, and pathological masses [7] or to predict surgical drainage [9]. Of late, magnetic resonance imaging (MRI) has been acknowledged as a feasible and highly accurate primary imaging method in the emergency setting in both the general population [10] and specifically in children [11]. The apparent advantages of MRI over CECT are its superior diagnostic accuracy [12] and lack of ionizing radiation. The distribution of deep neck infections and MRI findings differ between adults and children [11]. These differences are partly due to the varying proportional distribution and function of lymph nodes in different age groups [13]. Due to these differences, the interpretation of pediatric neck MRI studies requires specific knowledge and attention.

The purpose of the current pictorial review is to present typical neck infection MRI findings in children, including the most useful edema patterns and typical abscesses, and underline the potential caveats and complications associated with deep neck infections. We focus on the soft tissue of the neck above and below the hyoid bone. Pediatric non-traumatic emergencies of the orbits, nose, and ear have recently been covered elsewhere [14].

## 2. MRI Protocol and Practical Performance

We utilized the Philips Ingenia 3 Tesla scanner (Philips Healthcare, Best, The Netherlands). Our routine neck infection protocol comprises seven sequences and lasts approximately 30 min [15]. We use the same sequences for adults and children. The sequences are T2 Dixon (axial and coronal planes), T1 SE (axial), DWI (axial), and contrast-enhanced T1 Dixon sequences (axial, coronal, and sagittal) (Table 1). A gadolinium-based contrast agent (Dotarem^®^; Guerbet, Villepinte, France) is routinely used. T2-weighted imaging is useful in assessing the anatomy: fat and water have a bright signal, lymphoid tissue has an intermediate signal, muscles have a low signal, and cortical bone has no signal. Specific edema patterns can be appreciated as bright areas in the fat-suppressed, T2-weighted images. We prefer the Dixon method for fat suppression because it reliably yields homogenous images even with large fields of view. The T1-weighted axial sequence is used to reference post-contrast Dixon sequences and to detect bone marrow fat obliteration in odontogenic neck infections. Pre-contrast T1-weighted sequences are also useful in assessing postoperative hematomas and chronic fluid collections, which may have a high T1 signal intensity. Together with diffusion-weighted imaging (DWI), contrast-enhanced sequences are used for diagnosing and characterizing abscesses. These sequences are important for assessing cystic masses, tumors, and necrotic lymph nodes [15]. The careful assessment of potential lymph node pathology is critical, especially in children. The DWI sequence is produced using standard echo-planar imaging (EPI) with a b-value of 1000 s/mm^2^. In the context of neck infections, the DWI serves two functions. First, it is used for demonstrating or excluding diffusion restriction related to purulence in abscesses, and second, it is used for detecting lymphoid tissue in tonsils and lymph nodes. Our MRI protocol is consistent with the protocol suggested by a recent multicenter international consensus paper [16].

MRI has been proven feasible in pediatric patients with suspected neck infections [11]. In our previous validation cohort of 45 children, 16 (36%) were sedated with spontaneous breathing, and only 3 patients (7%) required general anesthesia [11]. The risk of ionizing radiation for CT and the potential adverse effects associated with sedation/anesthesia for MRI are frequently discussed in the literature [17]. Further, related to feasibility, MRI scanning can cause fear and anxiety in children which needs to be recognized and minimized. Potentially useful methods for mitigating anxiety include ambient surroundings, colors, music, and other audiovisual pastimes, a “scan buddy”, and a mini-sized toy scanner [18].

Image quality can be deteriorated by artifacts induced by patient motion or metallic foreign bodies, but the proportion of non-diagnostic MRI scans is low (about 1%) even in the acute setting [10]. Especially in pediatric patients, the dental hardware related to orthodontics can easily complicate the assessment of the DWI, hampering radiological diagnostics if odontogenic infections are suspected. Luckily, odontogenic infections are rare in small children [11].

## 3. Anatomical Considerations, MRI Terminology, and Edema Patters

### 3.1. Neck Anatomy and Key Areas of Scrutiny

The normal anatomy of the pediatric oropharynx is presented in Figure 1. MRI accurately demonstrates lymphoid tissue in the tonsils, which may be quite prominent in small children and teenagers. The differences between children and adults in proportional neck anatomy and disease processes also create the framework for image findings and interpretation [11]. These include the differing distribution of lymph nodes in the neck, as nodes located in the retropharyngeal space tend to be affected by the infectious processes more often in children than in adults [10,11]. The lymph nodes are normally larger in children than in adults as lymph nodes can undergo atrophy and diminish in physical activity [13] (Figure 2 and Figure 3). No new lymph nodes develop with age. Children’s lymph nodes, both in the retropharyngeal space and laterally, are challenged by antigen exposure and presentation for the first time in these individuals’ lives, and when the nodes encounter a new antigen, they can become enlarged. In adult life, when challenged with a specific antigen, the immunological memory generates an antibody without the need for a new recognition response [13].

Viral or bacterial infections are the most common causes of acute lymphadenopathy in children [19]. The lymph nodes in viral cervical lymphadenitis are often soft, small, bilateral, mobile, and non-tender, whereas in bacterial-associated lymphadenitis, the nodes are usually unilateral, tender, and of acute onset. In non-viral cases, an empiric oral antibiotic is often prescribed as early as possible [20]. Large, reddened, and worsening lymph nodes may require hospitalization for parenteral antibiotics and occasionally surgical removal. Cervical lymphadenitis caused by nontuberculous mycobacteria is relatively common in children and may require surgical intervention; however, treatment strategies vary between countries and institutions [21,22,23].

As lateral lymphadenitis is common in children, the careful analysis of the lymph nodes is essential in looking for suppurative lymphadenitis (intranodal abscess) which may require surgery or drainage [11]. Children also tend to have more retropharyngeal deep neck infections than adults; thus, scrutinizing the retropharyngeal space anatomy and disease processes is necessary [11]. In children, severe deep neck infections may spread caudally into the mediastinum. We encourage extending at least one axial pre-contrast T2-weighted Dixon sequence and one axial post-contrast T1-weighted Dixon sequence down to the level of lung hilum in order to determine or exclude disease processes such as anterior or posterior mediastinal edema, pleural fluid collections, and mediastinal abscesses.

The thymus can often be seen in children as a homogenous mass with smooth margins in the upper mediastinum (Figure 4) and should not be mistaken for pathology. The superior part of the thymus often extends cranially, immediately below the left thyroid gland. Another normal structure that is usually easily visible in children on fat-suppressed T2-weighted images is the thoracic duct (Figure 5).

### 3.2. Terminology of Pathology

In MRI, we define infection as (1) an abnormal, high signal indicating edema in fat-suppressed T2-weighted sequences or (2) a high signal in fat-suppressed post-contrast T1-weighted sequences indicating pathological tissue enhancement. Abscess MRI criteria include an abnormal isointense or hyperintense collection on T2-weighted sequences with low ADC values and no central enhancement, as well as enhancement surrounding this collection on T1-weighted Dixon post-contrast sequences [15]. In diagnosing abscesses, all sequences should be carefully scrutinized together because lesions that have a low ADC value may be interpreted as either suppurative fluid or solid tissue with high cellularity, depending on the pattern upon contrast enhancement.

### 3.3. Edema Patterns

Reactive, non-suppurative soft tissue edema is common in acute neck infections [15]. While areas of soft tissue edema do not necessarily contain abscesses or other targets for surgically treatable fluid collections, they indicate the severity of the infection [15].

In children, two of the most useful edema patterns are retropharyngeal edema (RPE) (Figure 6), and mediastinal edema (ME) (Figure 7). RPE is seen in about half of the patients with acute neck infections [24,25,26]. The retropharyngeal space is bordered by the buccopharyngeal fascia and the superior constrictor muscle anteriorly and the prevertebral fascia posteriorly. The alar fascia further divides this compartment into the “true” retropharyngeal space anteriorly and the “danger space” posteriorly. The distinction between the true retropharyngeal space and the so-called “danger space” cannot be made on radiological grounds; thus, the term “retropharyngeal” is radiologically sufficiently adequate to denote both anatomical entities. RPE is as common in children as in adults and is a significant predictor of the need for treatment in the intensive care unit (ICU) [26].

ME is seen in about one-quarter of patients with acute neck infections and can be divided into two categories: anterior and posterior. Anterior ME is commonly a continuum of edema from the visceral and/or anterior cervical spaces, whereas posterior ME is a continuation of RPE caudally. Similar to RPE, ME is encountered as often in children as in adults and is a significant predictor of the length of hospital stay [26].

## 4. Typical Pediatric Deep Neck Infections

### 4.1. Tonsillitis, Peritonsillar Abscesses, and Parapharyngeal Abscesses

Tonsillitis (pharyngotonsillitis) is a common oropharyngeal infection. In our previous validation cohort, these types of infections were proportionally less common in children than in adults [11]. MRI findings in tonsillitis include edema and the post-contrast enhancement of the palatine tonsils and surrounding oropharyngeal mucosa. Due to high cellularity, the tonsils display innately restricted diffusion (Figure 1). Post-contrast imaging sequences are beneficial in distinguishing intrinsic diffusion restriction from peritonsillar abscesses (PTA) that exhibit pathological diffusion restriction [15]. PTAs form in the oropharyngeal mucosal space, the potential space between the tonsillar capsule and the superior constrictor muscles [27] (Figure 8) and are the most common type of abscess found in many cohorts [28]. They are usually treated without imaging, and conservative treatment with antibiotics is sufficient. As local incision and drainage are usually not applicable due to lack of cooperation, immediate tonsillectomy in general anesthesia is a good treatment option. However, if a complicated course of illness is suspected, such as parapharyngeal swelling or an unsuccessful local incision in older children, patients may benefit from MRI from which the true extension of the PTA and possible complications can be determined. PTAs are frequently located superiorly or caudally from the craniocaudal midpoint and may thus be unreachable if simple ambulatory needle drainage is attempted. PTAs may be bilateral, and this can result in airway compromise. If the abscess breaches through the superior constrictor muscle and the buccopharyngeal fascia laterally, it may reach the parapharyngeal space [27] (Figure 9). Large parapharyngeal abscesses are likely to require surgical drainage or at least close surveillance [29].

### 4.2. Retropharyngeal Abscesses and Suppurative Lymphadenitis

Retropharyngeal infections tend to occur in young children [30]. In our previous validation cohort, all but one of the patients under the age of seven had a primary neck infection located in the retropharyngeal space or the lymph nodes; conversely, no patients aged 8–17 years had retropharyngeal infections [11]. Upon close inspection, most retropharyngeal infections actually represent suppurative lymphadenitis. They typically originate from the lateral retropharyngeal lymph nodes of Rouvière (Figure 10 and Figure 11) but can rarely originate from the midline nodes (Figure 12). Quite often, the lateral retropharyngeal nodal abscess extends laterally into the post-styloid parapharyngeal space (carotid space)—these two spaces communicate freely (Figure 10 and Figure 11). A “true” retropharyngeal abscess between the fascial planes also exists but is less commonly encountered [11] (Figure 13). In addition to suppurative lymphadenitis in the retropharyngeal space, lymph nodes containing purulence are found more laterally and superficially in the neck [11]. Lateral lymphadenitis may be diagnosed well with ultrasound. In these cases, MRI may be useful to confirm purulence (intranodal abscess formation) with DWI and to exclude any deep extension of abscesses. Ultrasound has been found to be an equally sensitive and specific method compared to CT in lateral neck abscesses in children [31]. Smaller children with *Staphylococcus aureus* infections tend to present with lateral lymphadenitis with large infectious mass-like lesions accompanied by widespread soft tissue edema, and careful assessment of both DWI and fat-saturated post-contrast T1-weighted sequences is crucial (Figure 14). Large abscesses and those not responding to medical treatment may require surgery, preferably via a transoral route, if the abscess does not extend lateral to the carotid arteries [32,33] (Figure 9, Figure 10 and Figure 11).

### 4.3. Oral Cavity

Teeth are the most common cause of infection in the oral cavity. Luckily, they are rare in children [11] but may be encountered in teenagers with tooth infections. In children with primary teeth and thus many unerupted permanent teeth, clinical diagnosis and exploration may be complicated because of honeycombed anatomy, especially of the mandible. Odontogenic neck infections can be accurately described using MRI [15]. The typical locations for odontogenic abscesses are the sublingual and submandibular spaces (Figure 15). In our clinical practice, we do not use the historical term “Ludwig’s angina” because it is vague and does not capture actionable imaging outcomes: etiology and surgically drainable abscesses.

### 4.4. Sialadenitis

Infections of the salivary glands (sialadenitis) in children may be bacterial or viral [34]. In acute sialadenitis, MRI demonstrates a swollen salivary gland with edematous surroundings (Figure 16 and Figure 17). Abscesses can be identified using DWI and post-contrast T1-weighted images. In obstructive sialadenitis, sialoliths may be seen as foci of the signal void [15].

## 5. Challenging MRI Findings and Pitfalls

### 5.1. Lymphadenitis with Purulence vs. Necrosis

In early lymphadenitis, the lymph node may be necrotic but not yet abscessed. It may be difficult to differentiate between a non-enhancing lymph node with restricted diffusion (a low ADC because of purulence, indicating intranodal abscess) and a lymph node with delayed enhancement accompanied by restricted diffusion (a low ADC because of high cellularity of lymphoid tissue, indicating necrotic or near-necrotic lymphadenitis) (Figure 18). Indeed, necrotic lymph nodes may enhance slower than normal nodes, and the enhancement may not be noticed if later scans (about 10 min after contrast administration) are not carefully scrutinized. Therefore, necrosis and slow enhancement may be confused for non-enhancement in early scans.

### 5.2. Cystic Masses

Cystic masses, such as branchial cleft cysts, thyroglossal duct cysts, and vascular malformations, comprise the majority of congenital neck masses in children [35]. Some cystic lesions, such as lymphatic malformations, can grow rapidly during an acute neck infection. In general, the secondary infection of a cystic mass is suggested by a thick enhancement of the cyst wall and edema of the surrounding soft tissues (Figure 19, Figure 20 and Figure 21). ADC maps from DWI are useful in detecting or ruling out purulence of the cyst fluid (Figure 20 and Figure 21).

### 5.3. Artifacts

Gas can be difficult to detect from MRI because the signal void may not be easily distinguished from other areas of low signal [15]. All sequences should be carefully evaluated for a complete and consistent signal void (Figure 22). Metallic braces can cause a significant signal loss in the oral cavity in children and teenagers (Figure 22).

## 6. Complications

### 6.1. Mediastinitis

A mediastinal extension is related to a more severe course of illness in both children and adults [15] (Figure 23). The infection usually extends caudally from a primary neck infection site via the retropharyngeal route [11]. In order to screen for mediastinal disease, at least fat-saturated T2-weighted and post-contrast T1-weighted sequences in the axial plane should be extended to the upper mediastinum. If the whole mediastinum is to be imaged, the MRI protocol needs to be adjusted accordingly because of potential motion and susceptibility artifacts. Sometimes, a CT study is performed to confirm mediastinal findings (Figure 23).

MRI findings of mediastinal extension include anterior or posterior edema and/or abscess formation and pleural fluid accumulation (Figure 23). Minor edema findings are likely reactive (Figure 7) and should not be diagnosed as actual evidence of infection spread. On the contrary, extensive edema and especially abscess formation warrants acute clinical assessment for operative treatment and intensive care [15].

### 6.2. Venous Thrombosis

When venous thrombosis in the context of deep neck infection is diagnosed, the eponym *Lemierre’s syndrome* has historically been used. With internal jugular vein thrombophlebitis, this syndrome also encompasses bacteremia (typically *Fusobacterium necrophorum*) secondary to current oropharyngeal infection, resulting in septic emboli [36,37]. Lemierre’s syndrome has been associated with multiorgan failure and increased mortality risk [36] and thus requires prompt action. Luckily, this state is rare, but a moderate increase in incidence has been noticed in the past decades [38]. The syndrome does not always appear in its classical form, and a variant course of illness has been described [39].

Diagnosing thrombophlebitis is not always simple in MRI, as the thrombus may not be separable from a flow void in T1-weighted sequences (Figure 24). The lack of a flow void can also be indicative, and post-contrast sequences must be closely evaluated. In this context, MR angiography or a CECT of the head, neck, and thorax are alternative strategies to MRI to exclude possible complications, such as intracranial venous sinus thrombosis. Compared to CECT, the MRI scanning of multiple body parts is time-consuming, but MRI is more accurate in evaluating intracranial and epidural pathology in this setting.

## 7. Conclusions

Emergency MRI has been proven to be a feasible imaging method in children with acute deep neck infections. MRI has superior diagnostic accuracy compared to CECT, encouraging more widespread use, especially in younger individuals. The differences between children and adults in anatomic proportions and the distribution of infectious diseases in the neck result in distinctive MRI findings. The subtleties in MRI findings of pediatric deep neck infections require careful assessment and knowledge.

To summarize the MRI findings: all acute neck infections cause varying degrees of edema, which is characterized as an abnormally high T2 signal on fat-suppressed images and enhancement shown as a high signal in fat-suppressed post-contrast T1-weighted sequences. For example, edema and post-contrast enhancement of the palatine tonsils and surrounding oropharyngeal mucosa are typical MRI findings in tonsillitis. Abscesses, such as those in peritonsillar, parapharyngeal, and retropharyngeal spaces, exhibit peripheral enhancement and a central, non-enhancing collection of pus, which can be effectively visualized with DWI as low ADC values. The location or extension of the abscess through various anatomical structures (e.g., parapharyngeal abscess extending from the peritonsillar space lateral to the superior constrictor muscle) can then be used for proper classification. Careful examination of all MRI sequences is important to avoid various pitfalls. For example, necrotic lymph nodes may be difficult to distinguish from purulent lymphadenitis. Early scans may confuse necrosis and delayed enhancement for non-enhancement, and both can show low ADC values, indicating high cellularity or purulence.

## Figures and Tables

**Figure 1 children-10-00967-f001:**
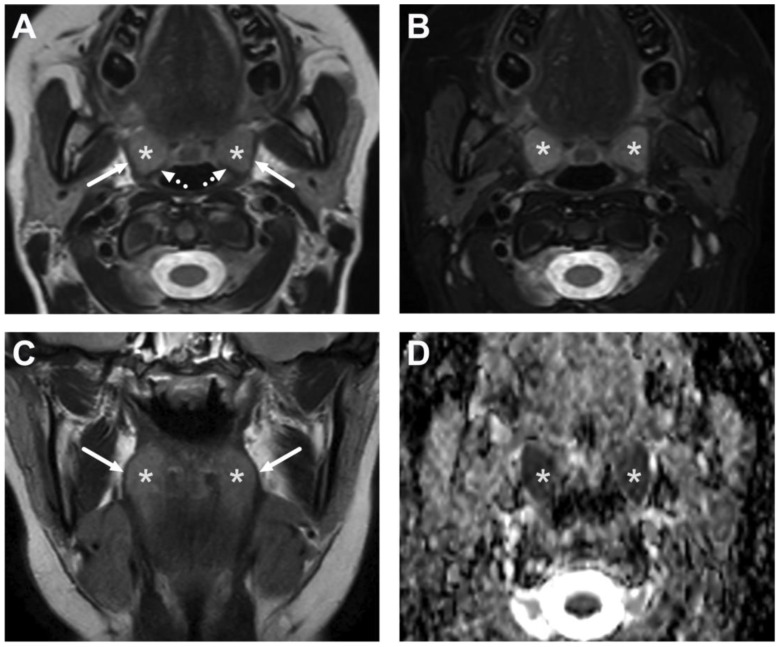
Normal MRI anatomy of the oropharynx in a 12-year-old. Axial T2-weighted (**A**), fat-suppressed T2-weighted (**B**), and coronal T2-weighted (**C**) images and an ADC map (**D**) demonstrate the palatine tonsils (white asterisk) bounded laterally by the superior constrictor muscle (arrows) and posteriorly by the palatopharyngeus muscle (dotted arrows).

**Figure 2 children-10-00967-f002:**
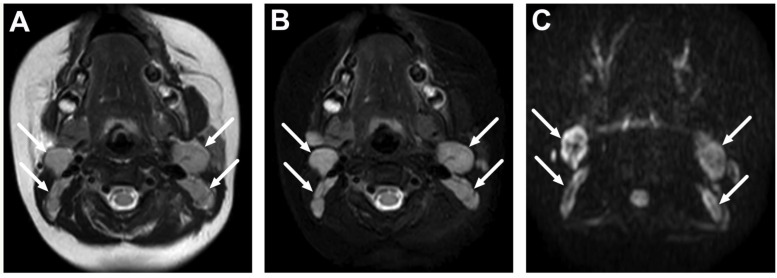
Normal level II lymph nodes in a 2-year-old on axial T2-weighted (**A**), fat-suppressed T2-weighted (**B**), and diffusion trace (**C**) images (arrows).

**Figure 3 children-10-00967-f003:**
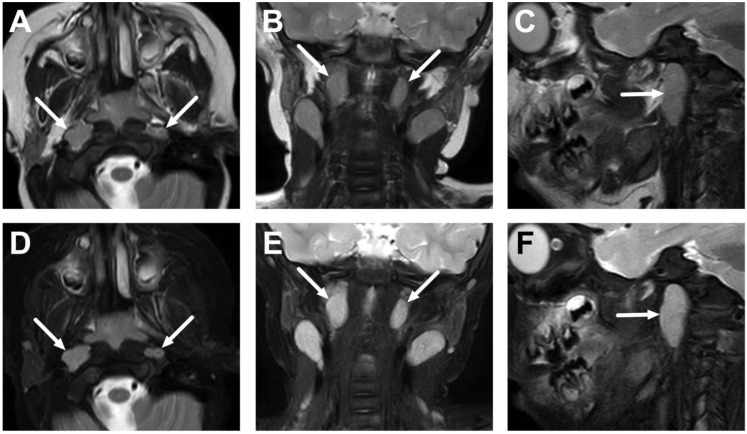
Normal lateral retropharyngeal lymph nodes (of Rouvière) in a 2-year-old on axial (**A**), coronal (**B**), and sagittal (**C**) T2-weighted images and axial (**D**), coronal (**E**), and sagittal (**F**) fat-suppressed T2-weighted images (arrows).

**Figure 4 children-10-00967-f004:**
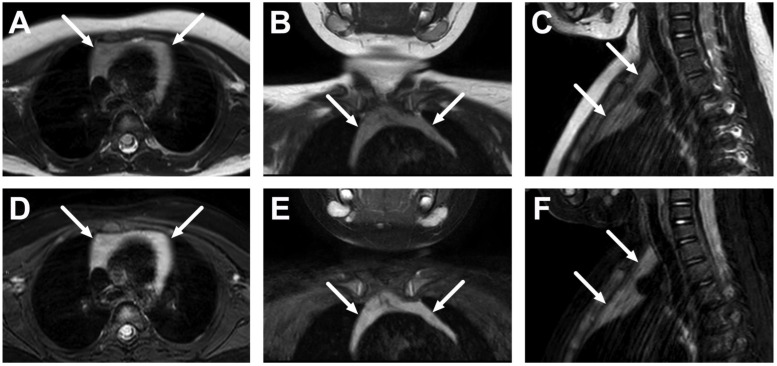
Normal thymus in the upper mediastinum in a 2-year-old on axial (**A**), coronal (**B**), and sagittal (**C**) T2-weighted images, and axial (**D**), coronal (**E**), and sagittal (**F**) fat-suppressed T2-weighted images (arrows).

**Figure 5 children-10-00967-f005:**
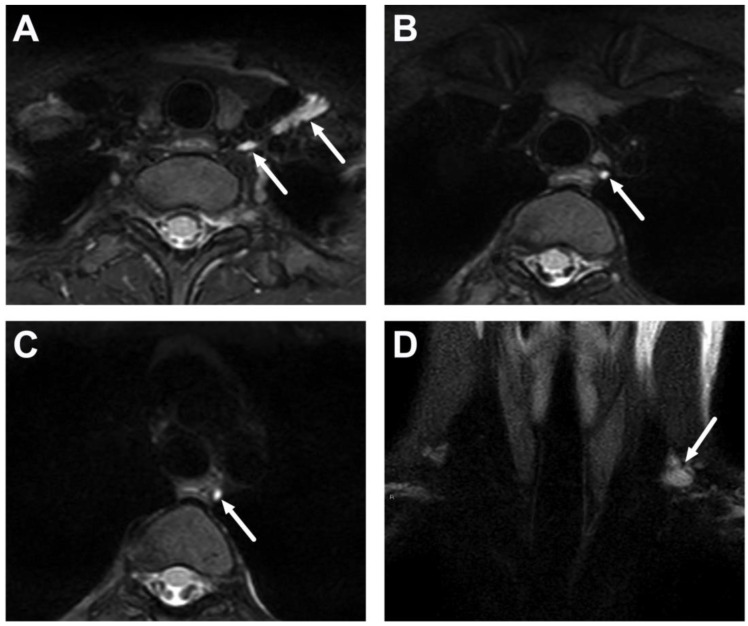
Normal thoracic duct (arrows) in a 15-year-old on three consecutive axial fat-suppressed T2-weighted images (**A**–**C**, superior to inferior) and a coronal fat-suppressed T2-weighted image (**D**).

**Figure 6 children-10-00967-f006:**
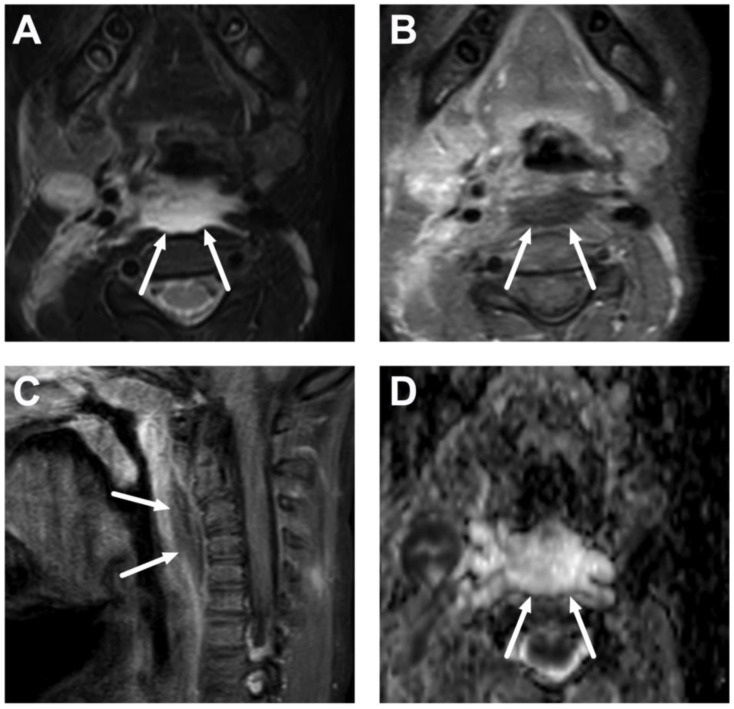
Reactive, non-suppurative edema (arrows) in the retropharyngeal space (RPE) in a 3-year-old child with lateral lymphadenitis. Images are axial fat-suppressed T2-weighted (**A**), fat-suppressed post-contrast axial (**B**), and sagittal (**C**) T1-weighted and an ADC map (**D**). The high signal in the ADC maps confirms that this non-enhancing fluid collection is not purulent but reactive. Surgical drainage is not required.

**Figure 7 children-10-00967-f007:**
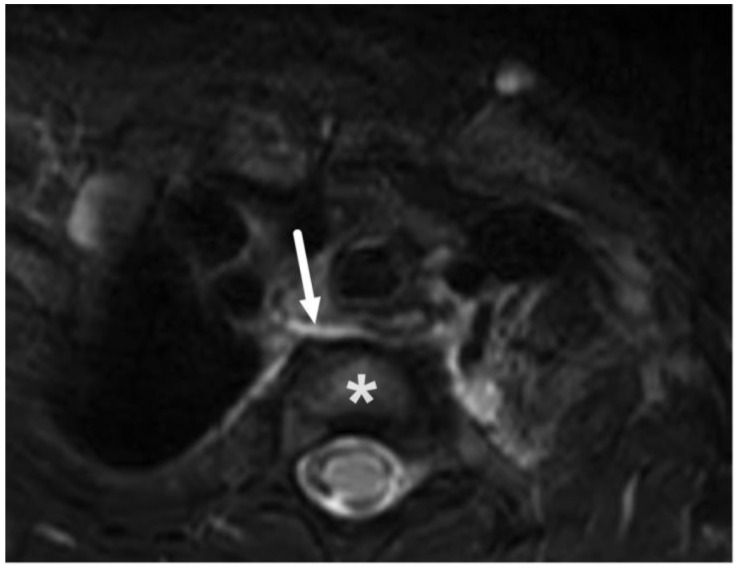
Reactive, non-suppurative edema in the mediastinum (ME) in a 3-year-old child with lateral lymphadenitis on an axial fat-suppressed T2-weighted image (arrow) at the level of the first thoracic vertebra (asterisk).

**Figure 8 children-10-00967-f008:**
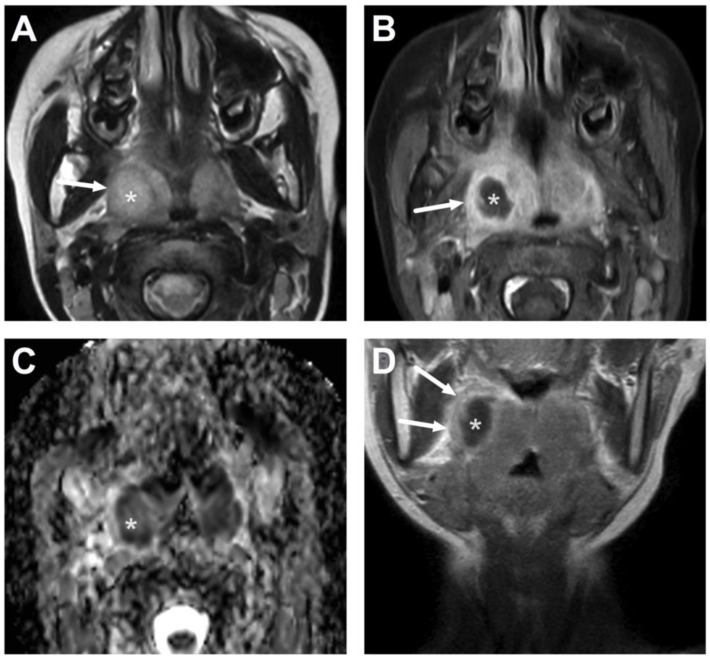
A peritonsillar abscess (PTA) (asterisk) in a 10-year-old on axial T2-weighted (**A**) and fat-suppressed post-contrast T1-weighted images (**B**), an ADC map (**C**), and a coronal post-contrast T1-weighted image (**D**). The abscess can be seen confined in the pharyngeal mucosal space, surrounded by the edematous superior constrictor muscle (arrows).

**Figure 9 children-10-00967-f009:**
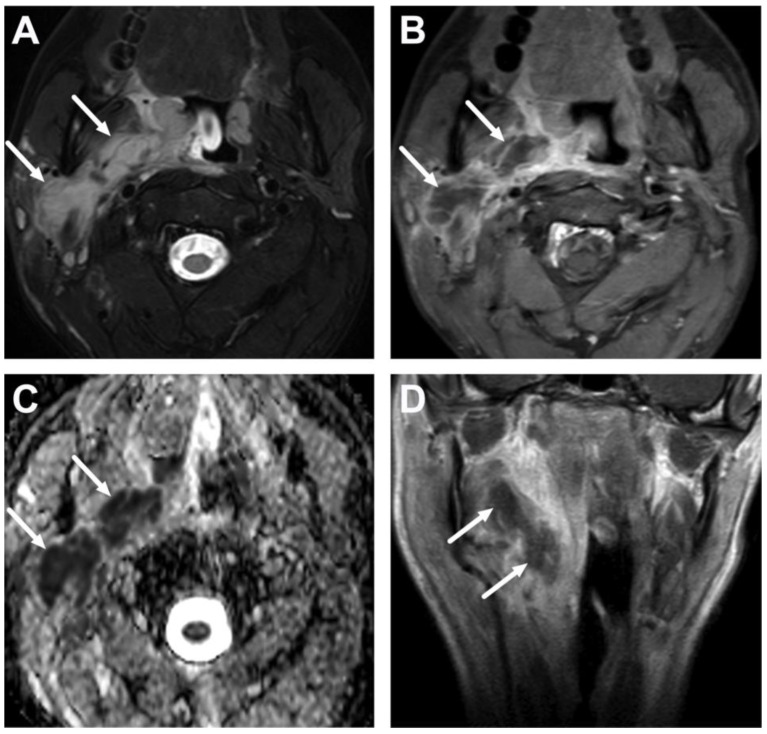
Parapharyngeal space abscess (arrows) in a 15-year-old teenager. Images are axial fat-suppressed T2-weighted (**A**) and fat-suppressed post-contrast axial T1-weighted (**B**) images, an ADC map (**C**), and a coronal post-contrast T1-weighted image (**D**). Notice that the abscess extends laterally, far beyond the border of the superior constrictor muscle. In image (**D**), the abscess can be seen extending caudally in the submandibular space, which communicates freely with the parapharyngeal space superiorly. Note that the abscess extends laterally beyond the internal carotid artery. Purulence was found during surgery, and *Fusobacterium necrophorum* was found in the pus culture.

**Figure 10 children-10-00967-f010:**
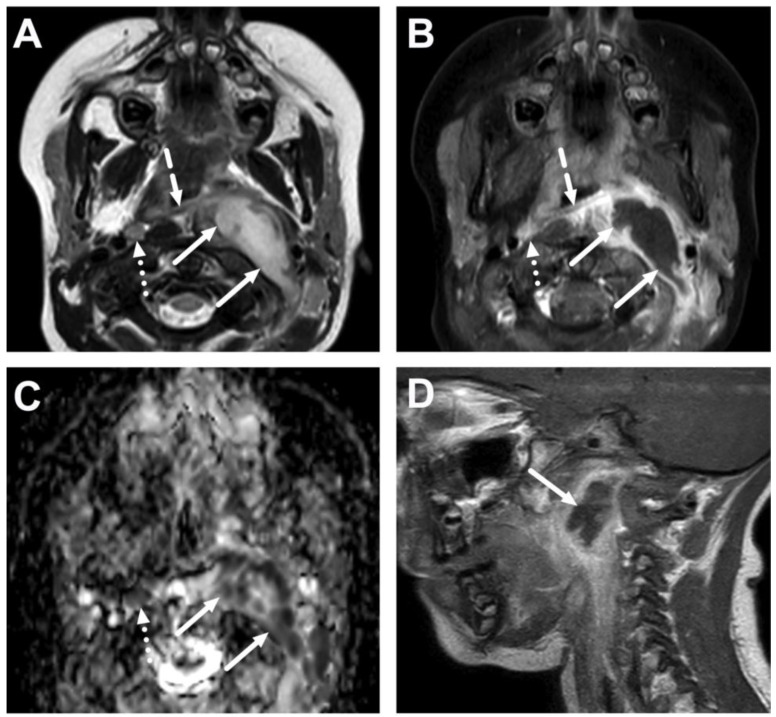
Suppurative lymphadenitis in the lateral retropharyngeal space in a 3-year-old child on axial T2-weighted (**A**) and fat-suppressed post-contrast T1-weighted images (**B**), an ADC map (**C**), and a sagittal post-contrast T1-weighted image (**D**). A large abscess (arrows) is posterior to the superior constrictor muscle (dashed arrows), confirming retropharyngeal location, and extends laterally to the post-styloid parapharyngeal (carotid) space. A normal retropharyngeal lymph node can be seen on the right side (dotted arrows). The abscess extends laterally slightly beyond the internal carotid artery. Purulence was found during surgery, and *Streptococcus pyogenes* was identified in the pus culture.

**Figure 11 children-10-00967-f011:**
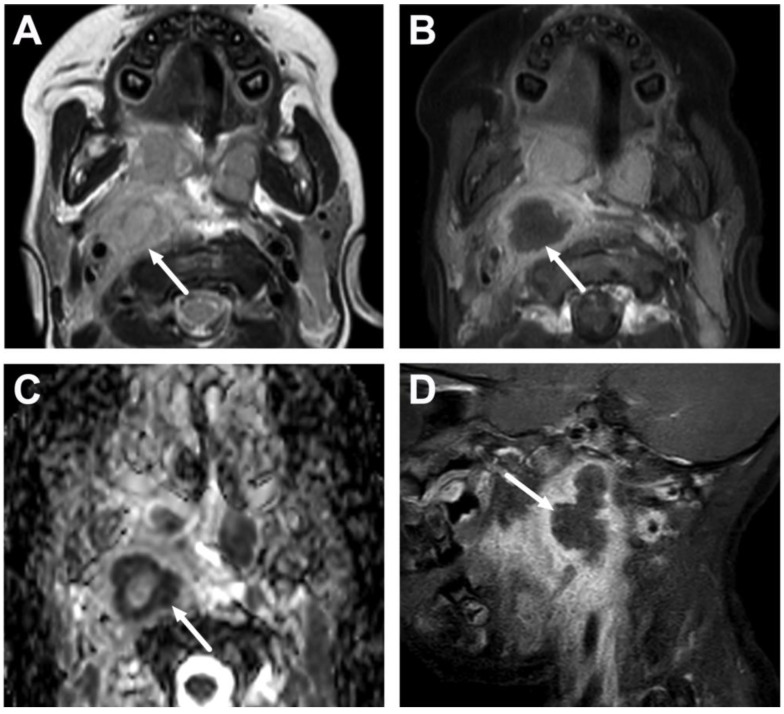
Suppurative lymphadenitis in the lateral retropharyngeal space in another 3-year-old child on axial T2-weighted (**A**) and fat-suppressed post-contrast T1-weighted images (**B**), an ADC map (**C**), and a sagittal fat-suppressed post-contrast T1-weighted image (**D**). A large abscess (arrows) extends laterally to the post-styloid parapharyngeal (carotid) space. Note that the abscess does not extend laterally beyond the internal carotid artery. Purulence was found during surgery, and *Streptococcus mitis* was identified in the pus culture.

**Figure 12 children-10-00967-f012:**
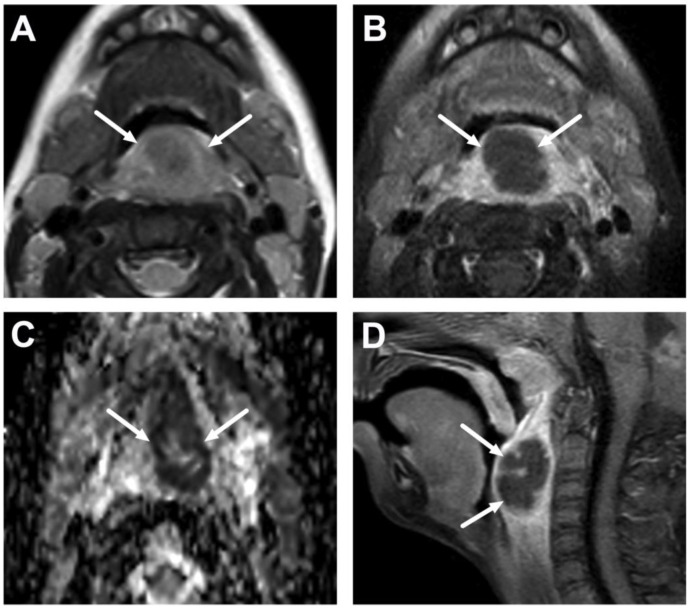
Suppurative lymphadenitis in the midline retropharyngeal space in another 3-year-old child on axial T2-weighted (**A**) and fat-suppressed post-contrast T1-weighted images (**B**), an ADC map (**C**), and a sagittal fat-suppressed post-contrast T1-weighted image (**D**). A large abscess (arrows) extends laterally to the post-styloid parapharyngeal (carotid) space. Purulence was found during transoral surgery.

**Figure 13 children-10-00967-f013:**
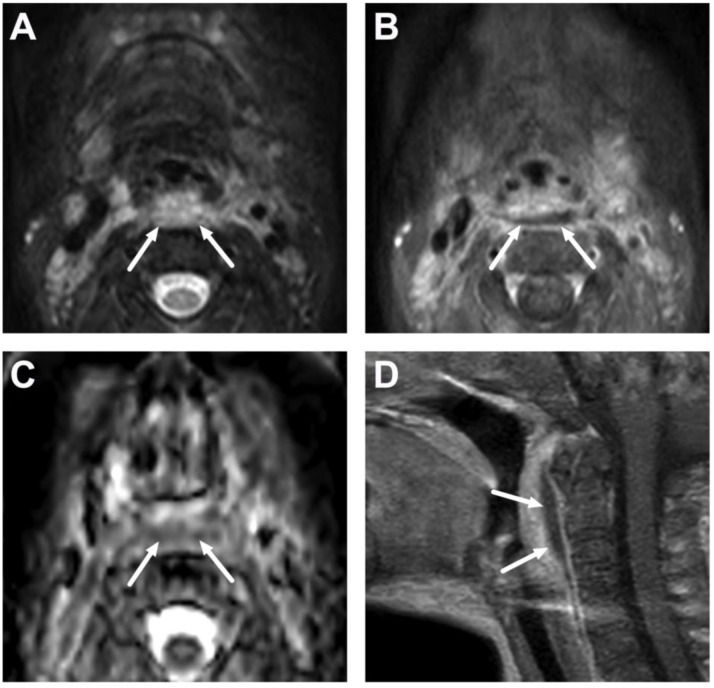
True retropharyngeal abscess in a 10-month-old infant on axial fat-suppressed T2-weighted (**A**) and fat-suppressed post-contrast axial T1-weighted (**B**) images, an ADC map (**C**), and a sagittal post-contrast T1-weighted image (**D**). A linear-shaped abscess (arrows) can be seen between the fasciae. Low ADC values (**C**) suggest purulence and not simply reactive edema.

**Figure 14 children-10-00967-f014:**
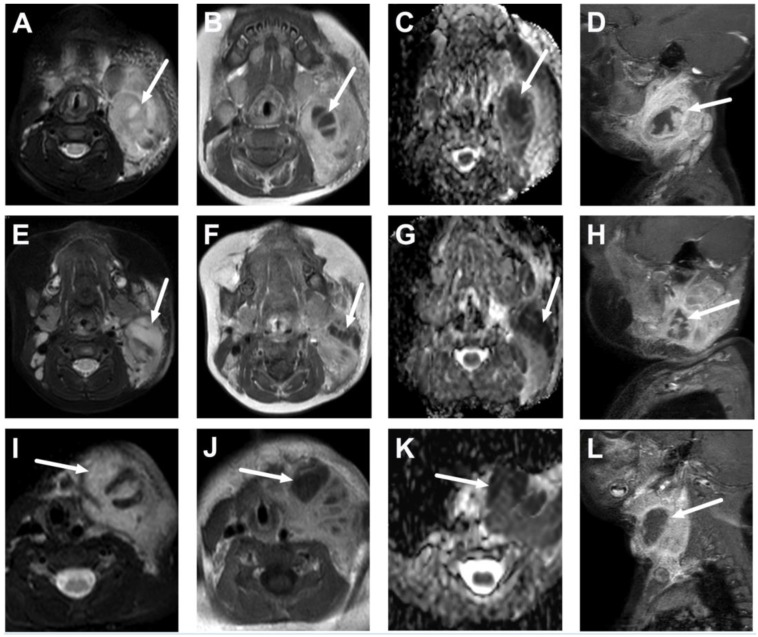
Lateral suppurative lymphadenitis in three infants less than 1 year of age (**A**–**L**). Images are axial fat-suppressed T2-weighted (**A**,**E**,**I**), post-contrast T1-weighted (**B**,**F**,**J**), ADC maps (**C**,**G**,**K**), and sagittal fat-suppressed T1-weighted (**D**,**H**,**L**). Note the large masses with surrounding edema and non-enhancing areas with low ADC values, which are indicative of intranodal abscesses (arrows). All patients underwent surgical drainage, and *Staphylococcus aureus* was found in the pus cultures.

**Figure 15 children-10-00967-f015:**
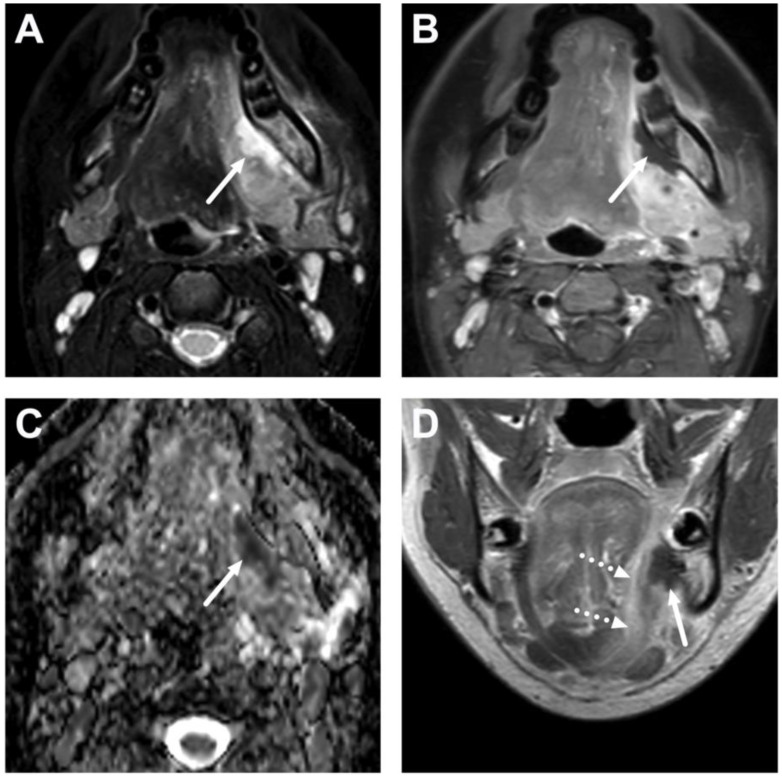
Odontogenic subperiosteal abscess (arrows) in the submandibular space following root canal treatment in a 15-year-old teenager. Images are axial fat-suppressed T2-weighted (**A**) and fat-suppressed post-contrast (**B**) images, an ACD map (**C**), and coronal post-contrast T1-weighted (**D**). An abscess is in the submandibular space, inferior to the edematous mylohyoid muscle (dotted arrows).

**Figure 16 children-10-00967-f016:**
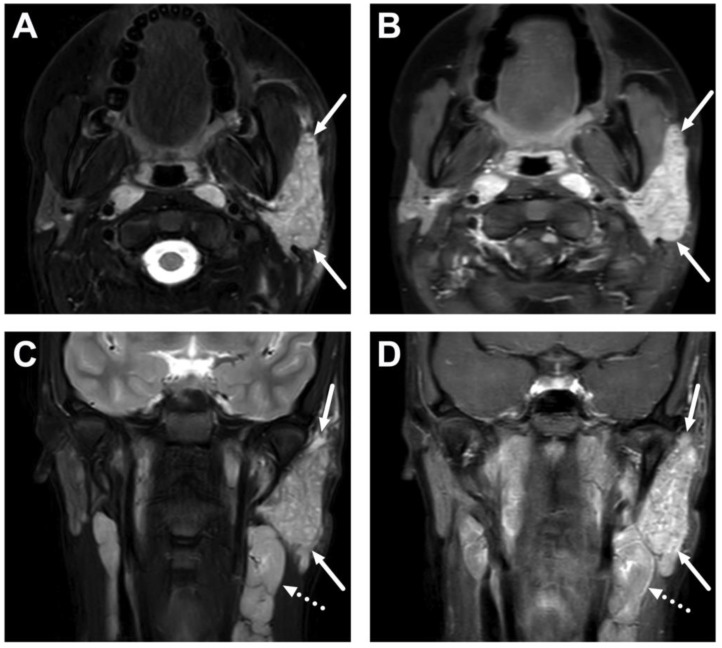
Acute parotitis of the left parotid gland (arrows) in a 16-year-old teenager. Images are axial (**A**) and coronal (**C**) fat-suppressed T2-weighted images and axial (**B**) and coronal (**D**) post-contrast T1-weighted images. The left parotid gland is swollen, edematous, and enhancing. No abscesses are seen. Note the slightly enlarged level II lymph nodes on the coronal images (dotted arrows).

**Figure 17 children-10-00967-f017:**
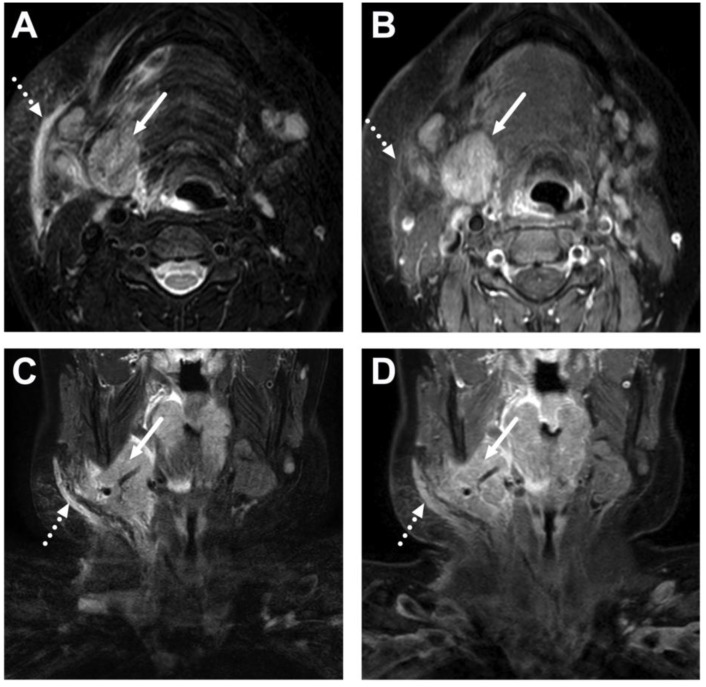
Acute sialadenitis of the right submandibular salivary gland (arrows) in a 13-year-old teenager. Images are axial (**A**) and coronal (**C**) fat-suppressed T2-weighted images and axial (**B**) and coronal (**D**) post-contrast T1-weighted images. The submandibular gland is swollen, edematous, and enhancing, but no abscesses exist. Note the considerable surrounding edema and enhancement in the adjacent soft tissues (dotted arrows).

**Figure 18 children-10-00967-f018:**
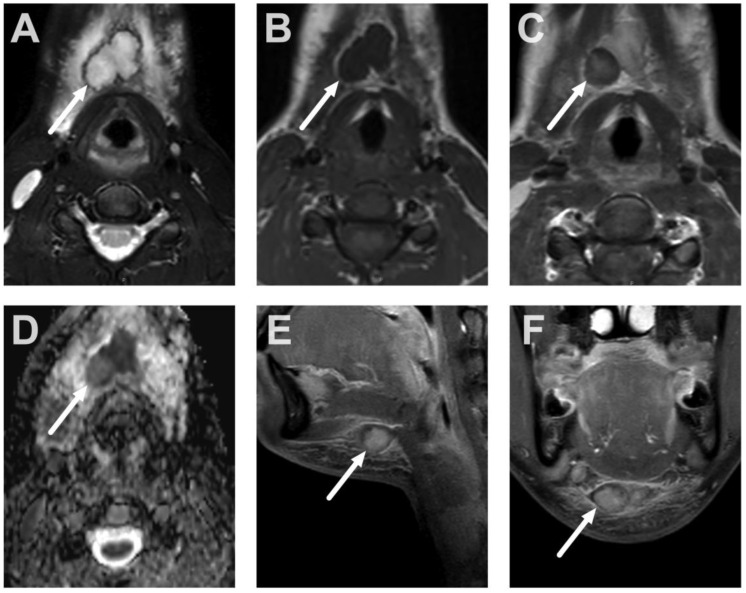
Necrotic lymphadenitis misinterpreted as suppurative lymphadenitis in a 15-year-old adolescent. Lymphadenitis is suggested on the axial fat-suppressed T2-weighted image (**A**) and poor enhancement on the post-contrast T1-weighted image (**C**) compared with the pre-contrast image (**B**) (arrows). The ADC demonstrates restricted diffusion (**D**) (arrow). The finding was initially misinterpreted as suppurative lymphadenitis (intranodal abscess formation); however, in post-contrast images taken at later time points, some delayed enhancement is seen (**E**,**F**) (arrows), ruling out suppuration. Consequent surgery found necrosis but no purulence. Image adapted from Ref. [15] under the Creative Commons Attribution License (CC BY 4.0).

**Figure 19 children-10-00967-f019:**
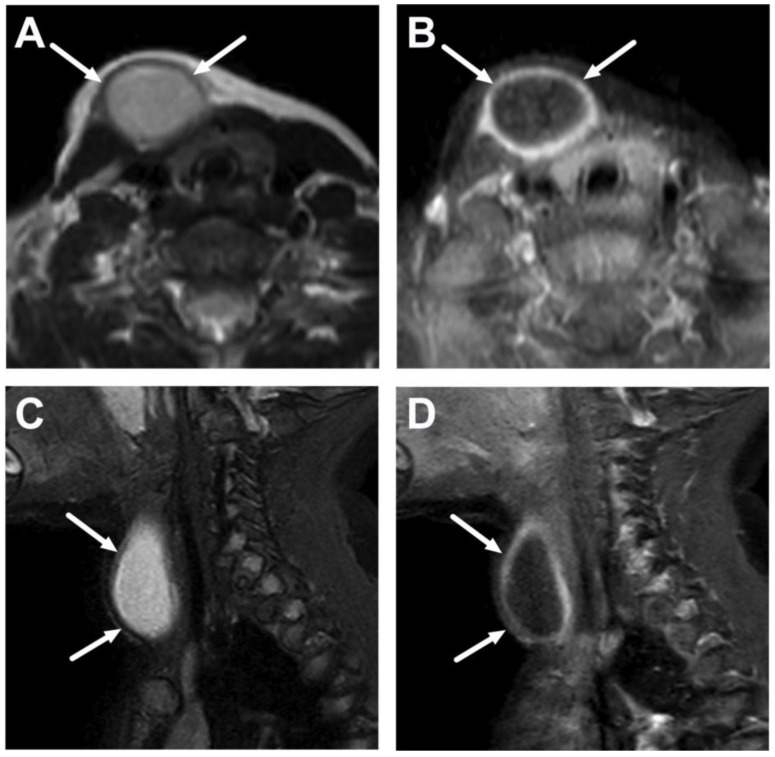
Second branchial cleft cyst (histologically confirmed) (arrows) in an 11-month-old infant on axial T2-weighted (**A**) and fat-suppressed post-contrast T1-weighted (**B**); and sagittal fat-suppressed T2-weighted (**C**), and post-contrast T1-weighted images (**D**). The thick rim enhancement suggests an acute infection; unfortunately, the ADC map was non-diagnostic in this case because of artifacts. The patient was managed medically in the acute phase.

**Figure 20 children-10-00967-f020:**
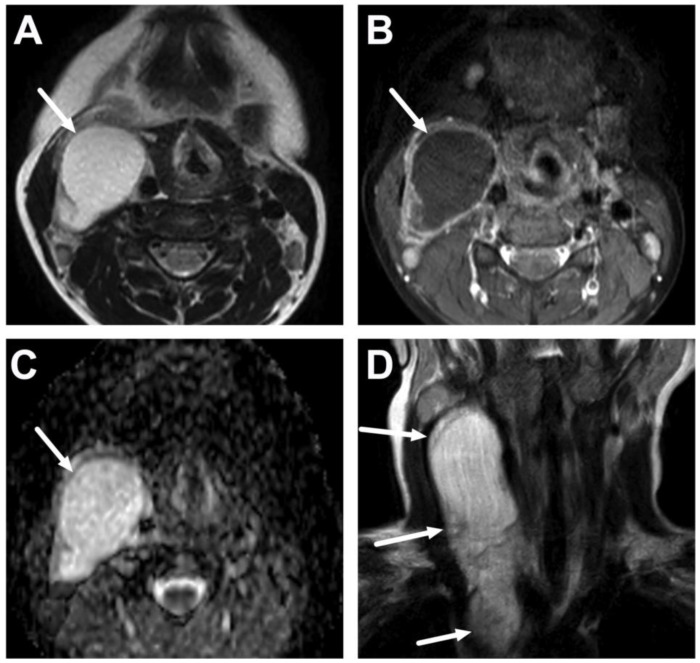
A large cystic mass (arrows) anteromedial to the sternocleidomastoid muscle in an 8-year-old child. Images are axial T2-weighted (**A**) and fat-suppressed post-contrast axial T1-weighted (**B**) images, an ADC map (**C**), and a coronal T2-weighted image (**D**). Acute enlargement and thick rim enhancement suggest an acute infection, but the ADC values are high, suggesting no purulence in the cyst. The patient was managed conservatively, and the cyst disappeared completely during follow-up, so no histopathological proof was obtained. The cyst extended caudally into the mediastinum (**D**), which is unlikely for a second branchial cleft cyst, suggesting the possibility of a thymic cyst, although these are rare and more commonly found on the left side.

**Figure 21 children-10-00967-f021:**
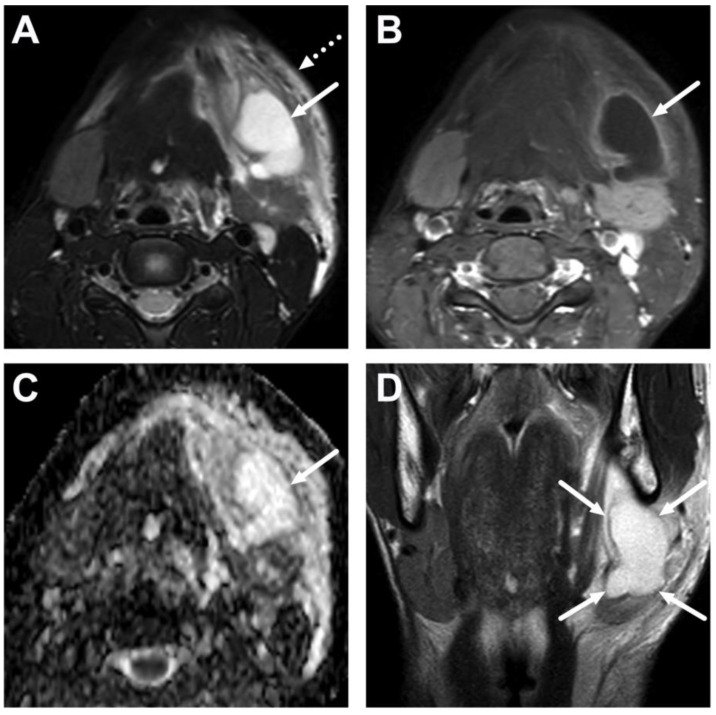
Lymphatic malformation with secondary infection in a 15-year-old teenager on axial T2-weighted (**A**) and fat-suppressed post-contrast axial T1-weighted (**B**) images, an ADC map (**C**), and a coronal T2-weighted image (**D**) (arrows). The coronal image (**D**) confirms the typical location in the submandibular space. Widespread edema of the surrounding soft tissues (dotted arrow in (**A**)) suggests an acute infection, but ADC values are high, suggesting no purulence in the malformation. The patient was later successfully treated with sclerotherapy.

**Figure 22 children-10-00967-f022:**
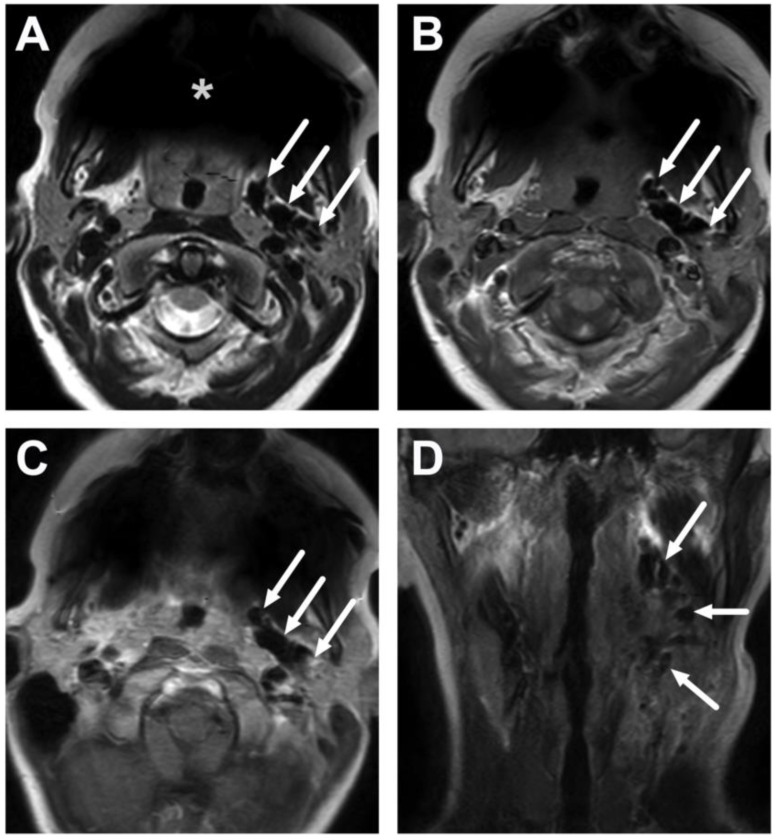
Neck pain and swelling after tonsillectomy in an 11-year-old child. Gas is shown as areas of signal void (arrows) in all sequences in the left parapharyngeal and submandibular spaces. Images are axial T2-weighted (**A**), pre-contrast T1-weighted (**B**), post-contrast T1-weighted (**C**) images, and a coronal post-contrast T1-weighted image (**D**). The anterior signal void (asterisk) is an artifact due to braces.

**Figure 23 children-10-00967-f023:**
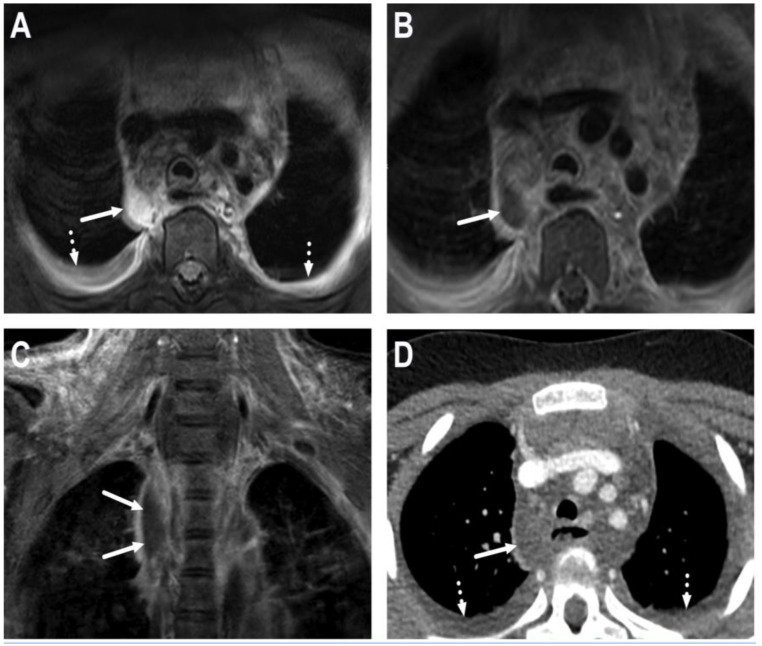
Descending mediastinitis in a 9-year-old child with widespread abscesses originating from the throat. The mediastinum was originally imaged with MRI; axial fat-suppressed T2-weighted (**A**) and post-contrast T1-weighted images (**B**); and a coronal post-contrast T1-weighted image (**C**). Imaging was consequently supplanted with CECT (**D**). Images show a mediastinal abscess (arrows) and pleural effusions (dotted arrows). The patient underwent surgery, and *Streptococcus pyogenes* was found in the pus cultures.

**Figure 24 children-10-00967-f024:**
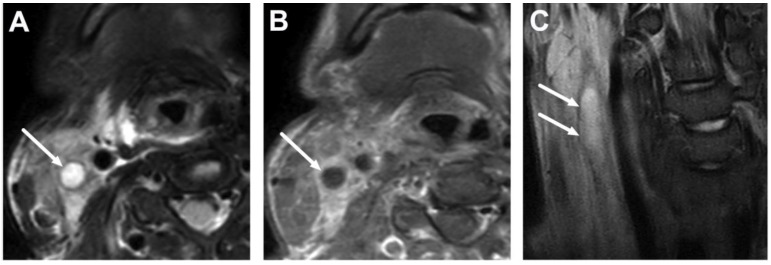
Thrombosed internal jugular vein (arrows) related to Lemierre’s syndrome in a 15-year-old adolescent with a throat infection and septicemia, demonstrated on axial fat-suppressed T2-weighted (**A**) and post-contrast T1-weighted (**B**) images, and a coronal T2-weighted image (**C**). Excluding thrombosis may be challenging on standard MRI sequences. In this case, the lack of flow void is seen on the T2-weighted images (**A**,**C**), whereas the hypointense thrombus is difficult to separate from the flow void on the T1-weighted image (**B**). *Fusobacterium necrophorum* was found in blood cultures. Image partially adapted from Ref. [15] under the Creative Commons Attribution License (CC BY 4.0).

**Table 1 children-10-00967-t001:** Practical approach to routine neck infection MRI protocol.

Phenomenon	Sequence	Findings	Suggestions	Notices
Soft tissue edema	T2 Dixon (water) post-contrast T1W Dixon (water)	Abnormal high signal	Radiologic evidence of an infection; specific edema patterns suggest a more severe course of disease.	All kinds of inflammation
Abscess	T1 SE T2 Dixon (water) DWI post-contrast T1 Dixon (water)	Non-enhancing collection with low ADC values enclosed in abnormally enhancing soft tissue edema.	Detection of an abscess usually requires operative consideration and exact abscess location, and extensions are useful in operative planning.	Abscesses may have an intermediate T2 signal content; blood products and/or postoperative status may complicate abscess assessment; necrotic lymph nodes may be misinterpreted as suppurative lymphadenitis.
Bone marrow edema	T1 SE T2 Dixon (water) post-contrast T1 Dixon (water)	Low signal in T1 and high signal in T2 Dixon (water) and post-contrast T1 Dixon.	Bone marrow edema may confirm odontogenic origin and point to the affected tooth.	Recent dental operations cause similar reactive findings; MRI artifacts may complicate assessment.
Complications	Whole protocol	Abscess extending to multiple deep neck spaces, mediastinis, venous thrombosis, and airway compromise.	Detection of potentially life-threatening conditions.	Magnetic resonance angiography (MRA) or CECT may be needed to diagnose venous thrombosis; defining airway compromise is difficult.
Cystic masses and potential neoplasms	Whole protocol	Identification of cystic component vs. neoplastic tissue, both with or without signs of infection.	Relevant differential diagnostics; exclusion of findings requiring immediate interventions.	Differential diagnosis may be limited and needs clinical correlation; biopsy may be required.

## Data Availability

Data availability is not applicable to this article as no new data were created or analyzed in the study.

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
