# Peer review of "Pictorial Review of MRI Findings in Acute Neck Infections in Children"

_children, 2023, doi:10.3390/children10060967_

Round 1

Reviewer 1 Report

Dear Authors,

Thank you for this excellent pictorial review comprehensively describing MRI findings of pediatric acute neck infections. I hope my input will be of use to the authors:

1) Briefly describe what constitutes pediatric acute neck infections in the introduction or methods, complete with imaging indications (What modalities should be used first, when is MRI indicated, etc.)

2) Some figures could use a better resolution, such as figures 9-13C, 15C, 17C and D.

3) Please add a conclusion where the authors summarize the findings of each disease entity in MRI.

4) This article has a 31% similarity report, with a 12% similarity report from this article (https://pubmed.ncbi.nlm.nih.gov/36617619/) which the authors also published. Furthermore, the authors self-cited seven (23%) of 30 references. I can accept up to 10-15% of self-citations, but almost a quarter seems excessive. Please justify this.

It is good enough to be understood

Author Response

Please, see attachment.

Reviewer 2 Report

"Cervical lymphadenitis caused by nontuberculous mycobacteria is relatively common in children, but it progresses slowly and rarely requires surgical intervention" is a wrong assertion because NTM cervical lymphadenitis should be surgically removed.

"Diagnosing thrombophlebitis is not always simple in MRI, as the thrombus may not be separable from a flow void in T1-weighted sequences (Fig. 24). The lack of flow void can also be indicative and post-contrast sequences must be closely evaluated. In practice, CECT of the head, neck, and thorax may be an alternative to MRI to exclude possible complications, such as intracranial venous sinus thrombosis", why not angioMRI instead of  CECT to see venous thrombosis?

I think that lateral neck lymphadenitis and cystic masses are sufficiently studied by US and doesn't need MRI except for exceptional cases

Author Response

Please, see attachment.
